# AudioMorphix: Training-free audio editing with diffusion probabilistic models

## Abstract

Despite recent advancements in diffusion-based audio generation, precisely editing content in a specific area of a recording remains challenging. In this paper, we introduce **AudioMorphix**, a training-free audio editor that manipulates a target area of a recording using another recording as a reference. Specifically, we conceptualize audio editing as part of a *morphing cycle*, in which different sounds can be combined into a cohesive audio mixture through morphing, whereas the mixture can be disentangled into individual components via demorphing. Leveraging the concept of audio morphing cycle, we optimize the noised latent conditioned on raw input together with reference audio and devise a series of energy functions to refine the guided diffusion process. Additionally, we manipulate the features within self-attention layers to preserve detailed characteristics from the original recordings. To accommodate a broad range of audio editing techniques, we collected a new evaluation dataset, providing editing instructions, reference audio and captions, and the duration of the edited area as guidance. Extensive experiments demonstrate that the AudioMorphix yields promising performance on various audio editing tasks, including addition, removal, and style transferring. Demo and code is available at this url.

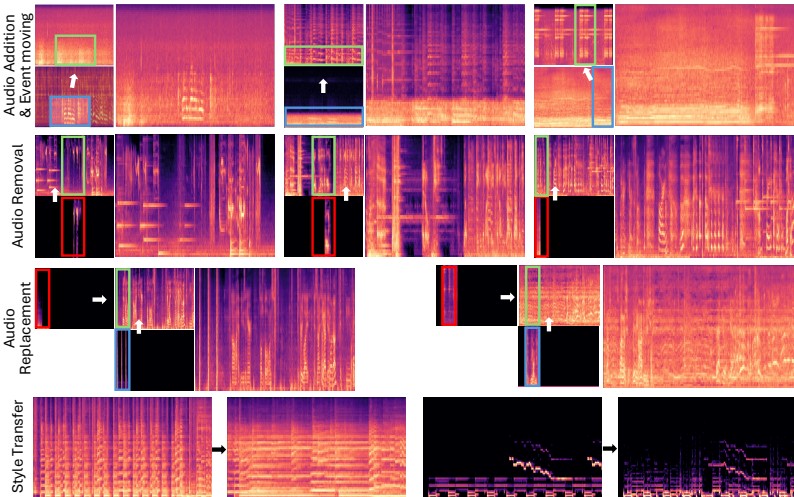

Figure 1: Audio editing tasks of which our AudioMorphix is capable with no training cost. We use green to highlight the editing region on the source audio while blue and red indicates the regions for addition and removal, respectively. The arrows represent the direction of the editing process, showing the flow from the reference audio to the source audio.

## 1 Introduction

Generative modelling has witnessed rapid breakthrough in the recent years, particularly in the domain of denoising diffusion models (Ho et al., 2020; Song et al., 2022). Despite progress was primarily seen

in the image synthesis, applications of audio generation have been attracting increasing interests (Liu et al., 2023; Ghosal et al., 2023; Majumder et al., 2024). While audio diffusion methods (Liu et al., 2023; Huang et al., 2023) are capable of generating diverse, high-fidelity audio, designing plausible guidance signals to create content consistent with user preference remains challenging.

Recent works have explored manipulating the audio content to align with user preference better. Wang et al. (2023) trained an end-to-end latent diffusion model to edit the audio content using text instructions. Manor & Michaeli (2024) and Liu et al. (2023) achieved editing by inverting raw audio into noisy latent and re-sample sounds from the obtained latent with new instructions. However, these audio editing methods heavily rely on textual instructions or description of the audio content and thus cannot precisely change the audio content within a specific region, limiting their ability to fully meet user preference. In the image domain, a local-editing approach was proposed by Shi et al. (2023) to alter the user-specified region in an image using a specified mask and a reference image. Nevertheless, such strategies cannot be directly applied to the audio domain: compared to images, where pixels can be simply added, audio mixtures often involves different sounds entangled in the same time-frequency (T-F) patches on the spectrogram. This naturally rises the question: *Can we edit sound in a controllable manner by guiding the diffusion process in the latent space?*

In this paper, we introduce AudioMorphix, a novel training-free sound editor that manipulates raw audio recordings conditioned on reference audio and a binary spectrogram mask. In particular, we cast audio editing as part of a *morphing cycle* performed on the latent space manifold (He et al., 2023; Yang et al., 2024): a sound mixture is created by morphing different recordings, while individual tracks are separated by demorphing the mixtures. Consequently, common audio editing tasks like addition and removal can be specified as latent morphing traversal on the manifold. By leveraging audio morphing cycle, we devise a gradient-based optimization method to iteratively search an appropriate noisy latent of the current diffusion process. We further design various energy function to guide the sampling process of latent diffusion models (Liu et al., 2023; Ghosal et al., 2023; Majumder et al., 2024). Additionally, we preserve the details of raw audio by substituting key, value components of self-attention layers in the current diffusion process with those from the reference audio, following empirical findings in the image domain. To evaluate a broader range of audio editing methods, as shown in Figure 1, we propose a new audio editing dataset that enables manipulation of raw audio content using various prompts, such as paired text description, task instruction, and reference audio.

Experiment results show that the proposed AudioMorphix outperforms state-of-the-art audio editing models on audio addition, removal, replacement and style transferring tasks. We also examine the impact of various system factors by ablation study of in our proposed components.

In summary, the contributions of this paper are as follows:

- We introduce a training-free framework that edits the specific region of raw audio by using pretrained audio diffusion models. To address the audio transparency, we cast editing tasks onto audio morphing cycle - a sound mixture is obtained by morphing two different sound while individual sounds are produced by demorphing a mixture.
- We proposed latent optimization method to estimate the targeted T-F patches of noisy latent via gradient-based optimization.
- We devise energy functions to guide audio generation along the trajectory of diffusion process. We also manipulate the features in self-attention layers by substituting the key and value components of the generation with those from the reference sounds.
- We create a new dataset to compare a broad range of audio editing methods by assessing the generated audio on particular T-F area. We show AudioMorphix outperforms current state-of-the-art methods on various tasks, including addition, removal, replacement, and style transferring.

## 2 PRELIMINARIES

### 2.1 DENOISING DIFFUSION MODELS

**Denoising diffusion models**. Diffusion models, or score-matching networks, have achieved great process in high-quality generation across various domains, such as image (Dhariwal & Nichol, 2021; Zhang et al., 2023), video (Xie et al., 2024), symbolic music (Zhang et al., 2024a) and audio generation (Liu et al., 2023). Let $x \in X \subset \mathbb{R}^d$ be a d-dimentional sample in the finite set of $X$,

drawn from the "true but unknown" distribution $P$, and $y \in Y$ be the provided condition, such as text description. Diffusion models generate a new sample by a sequence of invocation of time-dependent score function $\nabla_{x_t} \log p_t(x_t)$ for noisy data $x_t$. During training, a noise variable $\epsilon$ is sampled from Gaussian distribution $\epsilon \sim \mathcal{N}(0, \boldsymbol{I})$. The noisy data $x_t$ is obtained as a linear combination of the noise variable $\epsilon$ and the clean data $x \sim P(x)$ at the step $t$, as $x_t = \sqrt{\overline{\alpha}_t}x_0 + \sqrt{1-\overline{\alpha}_t}\epsilon$ where $\overline{\alpha}_t > 0$ is a scaling parameter. This conditional probability distribution can be defined by $q(x_t|x) := \mathcal{N}(x_t; \sqrt{\overline{\alpha}_t}x_0, (1-\overline{\alpha}_t)\boldsymbol{I})$. A diffusion model learns a denoiser $\epsilon_\theta(x_t, t)$ to parameterize the score function with the loss function

$$\mathbb{E}_{x_0, t, \epsilon_t \sim \mathcal{N}(0,1)} \left[ \|\epsilon_t - \epsilon_\theta(x_t, t)\|_2^2 \right], \tag{1}$$

where $\theta$ is a set of learnable parameters of the denoiser. In the sampling process, we apply the denoiser $\epsilon_\theta$ to estimate the noise variable $\epsilon_{t-1}$ and substitute it from noisy data $x_t$ iteratively to get the clean data $x_0$.

**Denoising diffusion implicit models (DDIM)**. DDIM was proposed to improve the inference speed by using deterministic generative process (Ho et al., 2020). During inference, DDIM obtains noisy data $x_{t-1}$ at the step $t$ with the following update rule:

$$x_{t-1} = \sqrt{\bar{\alpha}_{t-1}} \left( \frac{x_t - \sqrt{1-\bar{\alpha}_t}\epsilon_\theta(x_t, t)}{\sqrt{\bar{\alpha}_t}} \right) + \sqrt{1 - \bar{\alpha}_{t-1} - \sigma_t^2 \epsilon_\theta(x_t, t)} + \sigma_t \epsilon_t, \tag{2}$$

where on the right side the first term is an prediction of the clean data $x$ using the noisy data $x_t$ and the denoiser $\epsilon_\theta$, the second term represents the estimated dirction pointing to $x_t$, and the last term denotes a random noise. $\sigma_t$ is a scaling factor controlling the stochasticity in the sampling process: with $\sigma_t = \sqrt{(1-\bar{\alpha}_{t-1})/(1-\bar{\alpha}_t)}\sqrt{1-\bar{\alpha}_t/\bar{\alpha}_{t-1}}$. DDIM is implemented as DDPM while $\sigma_t = 0$ is interpreted as a deterministic sampling process. It is noteworthy that some works (Song et al., 2022; Lu et al., 2023) considered the deterministic sampling process as the discretization of a continuous-time probability flow ODE. This ODE-update rule can be reversed to give a deterministic connection between $x_0$ and its latent state $x_T$ (Ho et al., 2020), given by

$$\frac{x_{t+1}}{\sqrt{\beta_{t+1}}} - \frac{x_t}{\sqrt{\beta_t}} = \left( \sqrt{\frac{1-\beta_{t+1}}{\beta_{t+1}}} - \sqrt{\frac{1-\beta_t}{\beta_t}} \right) \epsilon_\theta^{(t)}(x_t). \tag{3}$$

For inference effiency, Salimans & Ho (2022) defined velocity $v$ as the combination of a clean sample $x_0$ and noise component $\epsilon$:

$$v_\phi = cos(\phi)\epsilon - \sin(\phi)x_0, \tag{4}$$

where $\phi_t = \arctan(\sigma_t/\alpha_t)$. Therefore, the DDIM sampling process can be re-wroten by:

$$z_{\phi_t} = \cos(\phi_t)x_0\epsilon(z\phi_t) + \sin(\phi_t)\hat{\epsilon}(z_{\phi_t}), \tag{5}$$

where $\hat{\epsilon}(z_\phi) = (z_\phi - \cos(\phi)\hat{x}_\theta(z_\phi))\sin(\phi)$. By applying the trigonometric identities, the update step can be written as

$$z_{\phi_t - \delta} = \cos(\delta)z_{\phi_t} - \sin(\delta)v_\phi(z_{\phi_t}). \tag{6}$$

**Classifier-free guidance (CFG)**. CFG is apllied to guidance the sampling process of diffusion models with an extra condition, such as text description. With CFG, a conditional and an unconditional diffusion model are jointly trained. At inference stage, the noise prediction can be obtained from conditional and unconditional estimates by

$$\epsilon_\theta(x_t, t, y) = w\epsilon_\theta(x_t, t, y) + (1-w)\epsilon_\theta(x_t, t, \varnothing), \tag{7}$$

where $w$ is guidance scale controlling the strength of condition signal on the generated output, and $\varnothing$ denotes the null token.

## 2.2 TRAINING-FREE GUIDANCE DIFFUSION

Recently training-free guidance diffusion methods are introduced to control the generated output by interfering the sampling process of diffusion models. In the community of images, DDIM inversion was proposed to manipulate an image by inverting it with the corresponding prompt and re-generate a new one conditioned on a reference prompt (Mokady et al., 2023). Prompt-to-prompt

framework (Hertz et al., 2022) was introduced to edit images by adjusting text description and attention map in cross-attention layers. Mokady et al. (2023) and Huberman-Spiegelglas et al. (2023) preserved in the diffusion process of source images the noise variable which is then used to adjust the noise variable of current images. Mou et al. (2024) and He et al. (2023) designed energy functions as an extra guidance on the top of noise estimation to control the sampling process of current images. In addition, some works (Mou et al., 2024; Chung et al., 2024) attempted to preserve the detailed information in source images by substituting the key, value vectors of the current sampling process with those of the source diffusion process. Despite the leap made in the image domain, there remains a non-trivial issue underlying in the community of audio: *sounds are transparent and always overlaps with each other*. In this work, we are studying on manipulating a sound track from sound mixtures while maintaining the rest of sound tracks in the audio.

## 2.3 Existing Works on Audio Editing

A straightforward approach for audio editing is to train a controllable audio generative model capable of taking extra condition as guidance. AudioBox (Team et al.), a flow-matching models conditioned on both text and audio prompt, was proposed to create the audio content by masking and audio infilling. Wang et al. (2023) and Han et al. (2023) trained dedicated diffusion model for various audio editing tasks, such addition, removal, replacement, and remixing. While these methods can be used for audio editing, large-scale training is required for a satisfying result, which could be impractical in some scenarios.

Some recent works focused on fine-tuning off-the-shelf models for audio editing (Wang et al., 2023). Lin et al. (2024) finetuned MusicGen (Copet et al., 2023) on multiple music editing task by introducing extra signals as guidance. Plitsis et al. (2023) investigated several image editing methods, such as DreamBooth (Ruiz et al., 2023) and Textual inversion (Gal et al., 2022), for audio personalization. Despite the training cost is minimum, they still need to tuning the model on task-specific datasets.

Zero-shot audio editing tasks were introduced by inversing diffusion process. Liu et al. (2023) firstly demonstrated the potential of text-to-audio diffusion models for editing tasks using DDIM inversion. More recently, Manor & Michaeli (2024) applied an edit-friendly DDPM latent space to edit the audio content by word swapping. However, such methods require precise text decription for transcription, limiting themselves from some editing use case.

## 3 Audio Latent Manipulation in the Morphing Cycle

### 3.1 Objective

In this work, we will focus on denoising diffusion models where the sampling process will be manipulated with reference audio. The proposed AudioMorhix features: (1) **Tuning-free**: The AudioMorphix is a zero-shot editing method that does not require extra training to fit task-specific data; (2) **Audio-referenced**: Instead of text instruction (Wang et al., 2023) which could be ambiguous in some use cases, the AudioMorphix takes an extra audio as reference for editing; (3) **Versatile**: the AudioMorphix is an universal framework capable of diverse editing tasks, including addition, removal, replacement, and style transferring; and (4) **Region-specific**: The AudioMorphix enables to edit a particular region of audio spectrogram while keeping the rest unchanged during editing.

Let $Enc(\cdot)$ be the transformation function mapping an input signal $x$ to latent state $z$ in the diffusion process. While previous methods (Zhang et al., 2024b; Chung et al., 2024) directly control the trajectory of generation process, empirically we found:

Assumption (Latent Spatial Consistency). *The spatial information of $x$ can be inferred from the latent representation $z$, such that:*

$$\text{Sim}(z_i, z_j) = \text{sim}(Enc(x_i), Enc(x_j)) \propto \text{Sim}(x_i, x_j) \tag{8}$$

This Assumption is in line with the finding in Yang et al. (2024). However, in contrast to visual modalities, manipulating the latent of a sound or a spectrogram is even harder: *Sound tracks are always entangled with each other in a mixture*, resulting in one pixel in a T-F spectrogram-like representation is correlated to more than one sound tracks.

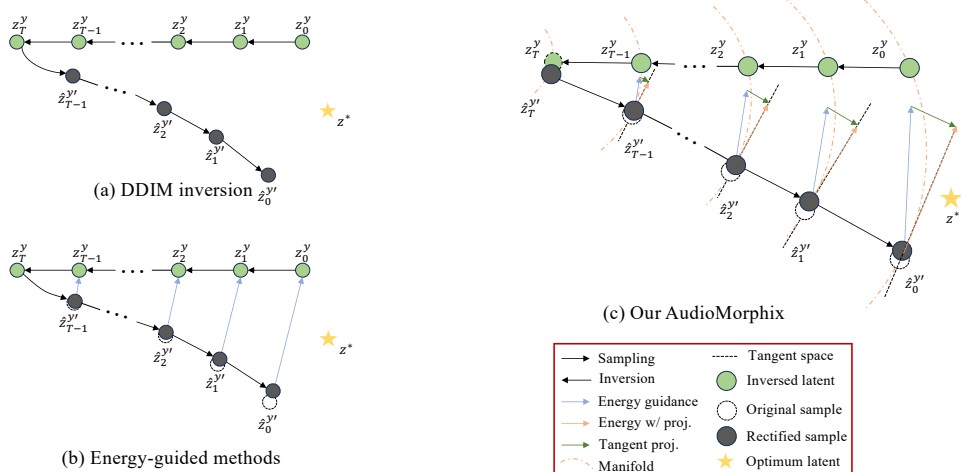

Figure 2: A schematic overview of our AudioMorphix in comparison with DDIM inversion (Mokady et al., 2023) and energy-guided methods (Mou et al., 2024). We omit the process of encoding input audio $x$ into latent $z_0^y$ for simplicity. AudioMorphix refines the sampling processing by updating the noisy latent $z_T^y$ and performing the energy guidance at each time step $t$. After obtaining the noisy latent $z_T^y$ with clean latent $z_0^y$ and the original text description $y$, AudioMorphix updates the noisy latent estimation $\hat{z}_T^{y'}$, where $y'$ is the reference text description, in the morphing cycle (in Section 3.2). Throughout the sampling process, AudioMorphix estimates the latent $z_{t-1}^{y'}$ via a trained latent diffusion model, guided by energy-based functions (in Section 4.2).

## 3.2 MANIPULATE LATENT IN THE MORPHING CYCLE

Let reference audio $x_r$ be the interested sound and context audio $x_c$ be the rest of sounds in the mixture. The mixture $x_m$ is the combination of reference sound $x_r$ and context $x_c$, such that $x_m = x_r + x_c$. According to our observation, the latent of the mixture $z_m$ also correlates with the latent of foreground and background sounds, $z_r$ and $z_c$, by: $z_m \propto z_r + z_c$.

Inspired by image morphing, we consider a mixture be an interpolation of the reference and context sound in the morphing path and reformat three basic audio editing operations from the perspective of the morphing cycle:

**Audio addition**: Provided an raw audio $x_c$ and a reference audio $x_r$, audio addition is to obtain the interpolation of the two sounds. He et al. (2023) and Yang et al. (2024) argued that latent states are distributed on a manifold, suggesting the infeasibility of linearly combining two latent states. Therefore, we interpolate between the latent state $z_c$ and $z_r$ [1] via spherical linear interpolation (SLERP) to obtain a "meaningful" intermediate latent state:

$$z_m = \frac{\sin((1-\alpha)\omega)}{\sin\omega} z_c + \frac{\sin(\alpha\omega)}{\sin\omega} z_r, \tag{9}$$

where $\omega$ is defined by $\omega = \arccos\left(z_c \cdot z_r / \|z_c\|\|z_r\|\right)$. The denoised result $z_m$ is then updated via the DDIM sampling by using the conditional distribution $p_\theta(x|y_c)$.

**Audio removal**: Audio removal is to separate a sound track $x_r$ from a mixture $x_m$ using audio $\tilde{x}_r$ as reference. Since the orthogonal directions of reference audio $\tilde{x}_r$ are not unique in a manifold, removing one sound with the reference audio *only* could result in satisfying editing result. To this end, we resort another sound track $\tilde{x}_c$ to regularize the sampling process of diffusion models. Algorithm 3.2 demonstrates how to optimize with gradient descent a latent state for the task of audio removal.

Instead of optimizing latent state $\tilde{z}_c$ and $\tilde{z}_r$ directly, the algorithm looks for the optimum direction pointing to $z_c$ and $z_r$. Because $z_c$ and $z_r$ are distributed on a sphere, we use SLERP and geodesic

---

[1]latent states hereby are referred to the noise latent at step T in the diffusion process. We ignore the subscription for simplicity.

**Algorithm 1** Latent optimization for the removal task
___
**Require:** $\tilde{z}_c, \tilde{z}_r\ z_m, t, lr, n_{iter}, use\_penalty, use\_tangent$
**Ensure:** Optimized neighborhood points $\hat{z}_c$ and $\hat{z}_r$
 1: Initialize $\epsilon_c \leftarrow \mathbf{0}, \epsilon_r \leftarrow \mathbf{0}$ with gradients
 2: $optimizer \leftarrow \text{SGD}([\epsilon_c, \epsilon_r], lr)$
 3: **for** $i = 1$ to $n_{iter}$ **do**
 4:     $optimizer.zero\_grad()$
 5:     $\hat{z}_c \leftarrow \tilde{z}_c + \epsilon_c$
 6:     $\hat{z}_r \leftarrow \tilde{z}_r + \epsilon_r$
 7:     $\hat{z}_m \leftarrow \text{SLERP}(t, \hat{z}_c, \hat{z}_r)$
 8:     $loss \leftarrow \text{GEODESIC\_DISTANCE}(z_m, \hat{z}_m)$
 9:     **if** $use\_penalty$ **then**
10:         $penalty \leftarrow (\sum \hat{z}_c \cdot \hat{z}_r)^2$
11:         $loss \leftarrow loss + penalty$
12:     **end if**
13:     $loss.backward()$
14:     **if** $use\_tangent$ **then**
15:         $\epsilon_c.grad \leftarrow \text{TAGENT\_PROJ}(\epsilon_c.grad, \hat{z}_c)$
16:         $\epsilon_r.grad \leftarrow \text{TAGENT\_PROJ}(\epsilon_r.grad, \hat{z}_r)$
17:     **end if**
18:     $\text{GRADIENT\_CLIP}([\epsilon_c, \epsilon_r])$
19:     $optimizer.step()$
20: **end for**
21: **return** $\tilde{z}_c + \epsilon_c,\ \tilde{z}_r + \epsilon_r$
___

distance to calculate the interpolation and similarity, respectively. Assuming $z^c$ and $z^r$ are independent with each other, we use the similarity between them as a penalty score to regularize the optimization process. We also attempt to project the optimization direction upon the sphere to ensure the updated latent states are "meaningful" following previous works (He et al., 2023). We empirically set number of iterations $n\_iters = 100$, learning rate $lr = 1e^{-4}$ and enable the use of penalty function and tangent space projection.

**Audio replacement**: Audio replacement is to replace a sound track $x_{rs}$ from a mixture $x_m$ with another audio $x_{rt}$. We decompose the task of audio replacement by separating audio $x_{rs}$ and adding audio $x_{rt}$ upon the mixture $x_m$. We used the same setting as audio addition and audio removal, respectively.

## 4  STEPWISE GUIDANCE IN SAMPLING PROCEDURE

### 4.1  OVERVIEW

This section introduces a stepwise guidance to control the generation procesdure using the updated audio latent in Section 3. Motivated by previous methods (Mou et al., 2024; He et al., 2023), our goal is to decompose a conditional score function $\nabla_{x_t} \log p(x_t|c, x^r)$ into a text-to-audio conditional score function and a differentiable term: $\nabla_{x_t} \log p(x_t|c, x^r) = \nabla_{x_t} \log p(x_t|c) + \nabla_{x_t} L_t(x_t; x^r)$. While there are some works devising energy functions for visual editing, we further improve them by considering latent in the diffusion procedure as T-F representation.

### 4.2  GUIDE AUDIO EDITING WITH ENERGY FUNCTION

In the AudioMorphix, various energy functions are devised as an extra guidance to control the audio generation procedure, mainly focusing on content consistency and contrast between generated audio and reference audio.

The first derivative of a energy function is added to the score obtained from the conditional U-Net $\epsilon_\theta$ for latent update in the sampling process. Suppose $F_t^c, F_t^r$ are the intermediate features obtained from the conditional U-Net $\epsilon_\theta$ at step $t$ corresponding to the input audio and the reference audio, respectively. Empirically, we collate the intermediate features $F_{t,l}^c, F_{t,l}^r$ from the $l$-th self-attention

layers of the U-Net decoder Let $m^c$ and $m^r$ be the binary masks upon the spectrogram of input audio and reference audio, respectively. The binary masks $m^c$ and $m^r$ can constrain the audio editing operating on particular T-F patches. We can measure the consistency between input and reference audio by calculating their cosine similarity over the interested area:

$$\text{sim}(F_{t,l}^c, m_c, F_{t,l}^r, m_r) = 0.5 \cdot \cos\left(F_{t,l}^c[m_c], s_g(F_{t,l}^r[m_r])\right) + 0.5, \tag{10}$$

where $sg(\cdot)$ is the gradient clipping function. Intuitively, we scale the similarity score $\text{sim}(\cdot) \in [0, 1]$ to align with human perception where $0$ means the closest distance between two audio. The guidance of consistency term is then defined by:

$$S_{\text{consist}}(F_t^c, m_c, F_t^r, m_r) = \sum_{l \in L} \frac{1}{1 + 4 \cdot \frac{1}{HW} \sum_{h \in H} \sum_{w \in W} \text{sim}(F_t^c, m_c, F_t^r, m_r)}, \tag{11}$$

While the contrast concept between two audio can be defined as the reciprocal of the cosine similarity, we argue that in the audio removal use case, the sound track in the reference audio is similarly but not the same as that of the input audio. Therefore, the contrast between input and reference audio is measured with the global representation of the input and the reference:

$$S_{\text{contrast}}(F_t^c, m_c, F_t^r, m_r) = \frac{1}{HW} \sum_{h \in H} \sum_{w \in W} \text{sim}(F_t^c, m_c, F_t^r, m_r), \tag{12}$$

Notably, the proposed energy functions is capable of generalizing to various prediction objectives, including DDIM and v-prediction, by directly modifying the probability density distribution to rectify the sampling trajectory. In experiments, we set $L = 2, 3$ be the selected self-attention layers of the U-Net decoder.

## 4.3 Energy guidance for Each Task

Exploiting the consistency measurement $S_{\text{consist}}$ and contrast measurement $S_{\text{contrast}}$, we devise a variety of energy-based function:

**Audio addition**. The goal of audio addition is to mix the context audio $x^c$ with the reference audio $x^r$. $m_c$ and $m_r$ are the binary masks of context and reference audio, respectively. Since sound tracks are "transparent", the original sounds in the context audio cannot be replaced with those of reference audio. Therefore, the devised energy function should consider not only the consistency between reference and generated audio, but also the consistency before and after edition. The energy-based guidance can be expressed in the follow:

$$\epsilon_{add} = w_{content} \cdot S_{\text{consist}}(F_t, m_c, F_t^c, m_c) + w_{edit} \cdot S_{\text{consist}}(F_t, m_c, F_t^r, m_r). \tag{13}$$

**Audio removal**. Audio removal is to separate a sound track from the input mixture while preserving the rest of sounds. In addition to push the generated audio away from the reference audio in the latent space, we should keep the similarly of global representation unchanged such that:

$$\epsilon_{remove} = w_{content} \cdot S_{\text{consist}}(F_t, m_c, F_t^c, m_c) + w_{edit} \cdot S_{\text{contrast}}(F_t, m_c, F_t^r, m_r). \tag{14}$$

**Audio replacement**. We deem the replacement task as a chain of basic operations. Particularly, we exert removal and addition tasks separately to replace a sound track in the mixture with another one.

## 4.4 Diffusion procedure with Memory Bank

The combination of latent morphing and energy guidance build a good posterior in the diffusion sampling process. However, as some works indicates, the gap between generated and reference audio still exist. Following Mou et al. (2024), we modify the self-attention mechanism in the conditional U-Net. As shown in Figure 4.4, the key, value of self-attention layers in the decoder are substitute by the original ones obtained from the inversion process. In experiments, we replace the key, value of the second and the third layers with those of the inverted trajectory.

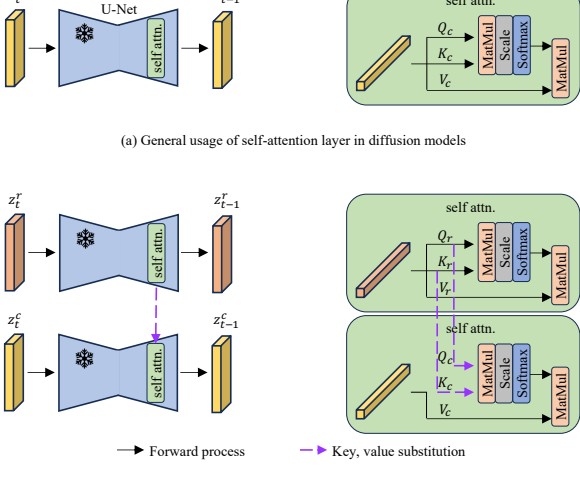

(a) General usage of self-attention layer in diffusion models

(b) Our implementation of substituting key and value in self-attention layer

Figure 3: Illustration of adapting self-attention layers to preserve detailed information in the reference latent $z_r^t$. We cache the key, value of reference latent $z_r^t$ and substitute those of latent $z_c^t$ during forward process.

## 5 EXPERIMENTS

### 5.1 EXPERIMENT SETUP

**Datasets**. To evaluate diverse editing methods, we curated a new dataset *AudioSet-E* based upon the temporally-strong labelled part of AudioSet (Gemmeke et al., 2017) for three audio editing tasks, including addition, removal, and replacement. AudioSet-E contains multiple options, including instruction, audio, pairs of text description, as reference for audio editing. Particularly, AudioSet-E contains 1442 samples for audio addition, 1426 samples for audio removal, and 1870 samples for audio replacement. See more about data curation in the Appendix A. In addition to AudioSet-E, we also evaluated the proposed AudioMorphix on style transferring. We applied MusicDelta, following the previous work (Manor & Michaeli, 2024).

**Comparison methods**. For addition, removal, replacement tasks, we compared our AudioMorphix against DDIM inversion (Liu et al., 2023), DDPM inversion (Manor & Michaeli, 2024), and AUDIT (Wang et al., 2023) on the AudioSet-E. We didn't implement DreamBooth and text inversion methods from Plitsis et al. (2023) because they are targeted at audio personalization rather than manipulation. Following Manor & Michaeli (2024), we also implemented AudioMorphix together with DDIM and DDPM inversion on style transferring. In addition to orginal audio, DDIM and DDPM inversion take a pair of original and target text descriptions as input while for AUDIT an editing instruction are required.

**Metrics**. We applied Frechet audio distance (FAD), kullback–leibler divergence (KL), Inception Score (ISc), CLAP score and LPAPS. to evaluate all audio editing model. FAD measures the fidelity between generated samples and target samples. KL measures the correlation between generated samples and target samples. ISc measures the diversity of generated audio. CLAP score measures the correlation between generated samples and target text descriptions. LPAPS calculates the perception distance between two audio using deep neural network. We release our evaluation kit [2] to facilitate a fair comparison in the future work.

### 5.2 COMPARISONS

Table 1 compares various audio editing methdos on the AudioSet-E evaluation dataset. Our AudioMorphix outperforms the comparison methods across all tasks, particularly excelling in terms of FAD and KL metrics, which indicates better fidelity and distribution matching of the edited images.

---

[2]https://anonymous.4open.science/r/TAGE-F1A8

Table 1: Comparison of various audio editing methods on the AudioSet-E evaluation set.

| | Addition | | | Removal | | | Replacement | | |
|---|---|---|---|---|---|---|---|---|---|
| | FAD ↓ | ISc ↑ | KL ↓ | FAD ↓ | ISc ↑ | KL ↓ | FAD ↓ | ISc ↑ | KL ↓ |
| DDIM inversion | 5.61 | **6.39** | 1.72 | 6.24 | **6.56** | 1.86 | 8.29 | **5.52** | 2.05 |
| DDPM inversion | 19.18 | 4.03 | 2.27 | 19.14 | 4.61 | 2.30 | 21.25 | 3.85 | 2.30 |
| AUDIT | 5.81 | 4.27 | 3.17 | 3.47 | 3.63 | 3.48 | 5.68 | 4.16 | 2.81 |
| Our method (w/ AudioLDM) | **5.58** | 4.43 | 0.83 | **2.83** | 4.19 | 1.29 | **2.67** | 5.05 | 2.28 |
| Our method (w/ Tango) | 6.62 | 5.26 | **0.57** | 6.29 | 5.11 | **0.77** | 7.27 | 4.28 | **0.62** |

This suggests that AudioMorphix provides more accurate and realistic image edits compared to DDIM inversion, DDPM inversion, and AUDIT methods. The results are especially notable in the Addition and Removal tasks, where it shows significant improvements in the KL divergence, indicating a more precise alignment with the target distribution.

Table 2: Comparison of various audio editing methods on the MusicDelta evaluation set.

| | CLAP ↑ | LPAPS ↓ |
|---|---|---|
| DDIM inversion | **0.32** | 6.80 |
| DDPM inversion | 0.29 | 7.17 |
| Our method | 0.31 | **6.75** |

Table 2 shows the experiment results of the proposed AudioMorphix and two inversion methods on a style transferring dataset (Manor & Michaeli, 2024). Our AudioMorphix yielded a better score than DDPM inversion. Compared to DDIM inversion, the proposed methods yielded a comparable CLAP score of 0.32 and a lower LPAPS score of 6.75. This observation aligns with our expectation: our AudioMorphix preserves the details of raw sounds while less aligned with text instruction as a trade-off.

## 5.3 ABLATION STUDY

Table 3: Ablation study on the choices of tangent space projection and text description.

| w/ Text | Tan. proj. | Addition | | | Removal | | | Replacement | | |
|---|---|---|---|---|---|---|---|---|---|---|
| | | FAD ↓ | ISc ↑ | KL ↓ | FAD ↓ | ISc ↑ | KL ↓ | FAD ↓ | ISc ↑ | KL ↓ |
| | | 6.46 | 4.10 | 0.84 | **2.46** | 4.69 | **1.03** | 6.06 | 3.12 | **1.00** |
| ✓ | | **5.58** | 4.43 | **0.83** | 2.83 | 4.19 | 1.29 | **2.67** | **5.05** | 2.28 |
| | ✓ | 8.49 | 1.73 | 3.08 | 3.08 | **5.40** | 1.63 | 8.09 | 3.00 | 1.06 |
| ✓ | ✓ | 6.10 | 4.70 | 1.50 | 3.38 | 5.08 | **1.03** | 5.91 | 4.10 | 1.70 |

We evaluated the components of AudioMorphix by ablating text description and tangent space projection in the Table 3. It can be observed that text description only significantly enhance the performance of the proposed method, achieving a FAD score of 5.58 and KL score of 0.83 on the addition task and a FAD score of 2.67 and an ISc score of 5.05 on the replacement task. Conversely, introducing tangent space projection led to performance degradation, especially on the addition and replacement tasks. This is likely because tangent space projection requires more steps for update compared to direct guidance.

## 6 CONCLUSION

In this paper, we proposed AudioMorphix, a training-free sound editor to manipulate raw audio conditioned on reference audio and binary masks. By casting audio editing as part of a morphing cycle performed on the latent space manifold, out approach promises high-fidelity audio editing within a controlable window, while without introducing training cost, paving the way for more controllable audio editing. This approach leverages the gradient-based optimization to iteratively search the appropriate noisy latent of the current diffusion process. The proposed method incorporates various energy functions that rectify the trajectory of the sampling process. Furthermore, AudioMorphix adopts key-value substitution within self-attention layers, preserving the details of raw audio during editing. The experiments on various audio editing tasks show the effectiveness and promise of AudioMorphix compared to previous audio editing methods.

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

APPENDIX

## A    DATASET CURATION

We curated a new dataset to evaluate various audio editing tasks, including addition, removal, and replacement, based on the temporally-strong subset of the AudioSet dataset (AudioSet-SL) (Gemmeke et al., 2017). Utilizing the timestamps of sound events in AudioSet-SL, we mixed 2-3 audio tracks together with or without the selected sound events. A separate dataset was created for each task as described below:

**Audio Addition**. We randomly selected a sound event from two audio samples in the database and created two mixtures: one with and one without the selected sound event. The mixture without the selected sound event was used as the raw audio, and the mixture with the event was used as the target audio. Additionally, we used the isolated sound event as the reference audio. For text descriptions, we used a bag of sound event categories from AudioSet-SL, filling predefined templates with the selected sound event's name as the instruction.

**Audio Removal**. The curation of the audio removal dataset follows a similar process to the audio addition task. However, the mixture with the selected sound events in the audio removal was used as the raw audio, and the mixture without those events served as the target audio. To increase the difficulty of the audio-driven editing task, we randomly sampled 1-second clips from the selected events and discarded the remaining portions during preprocessing.

**Audio Replacement**. We randomly selected three audio recordings, labeled A, B, and C, from AudioSet-SL. We ensured that A and B contained overlapping sound events from different categories. For the raw audio, we mixed audio C and the overlapped region from audio A, and for the target audio, we blended audio C and the same region from audio B. Recordings from A and B were used as the reference audio. For text descriptions, we used combinations of sound events from the two tracks (A and B), filling predefined templates with the relevant sound events as the editing instruction.

The resulting dataset, AudioSet-E, contains 1,442 samples for audio addition, 1,426 samples for audio removal, and 1,870 samples for audio replacement. Compared to previous audio editing datasets (Gui et al., 2024; Liang et al., 2024), AudioSet-E provides a more diverse platform to evaluate the quality of generated audio across multiple editing tasks.

## B    IMPLEMENT DETAILS

We used a single NVIDIA A100 for evaluation. For a fair comparison, our AudioMorphix was provided with no masking information same as the other editing methods. We set guidance scale 1 for AudioLDM and 1.2 for Tango. For our AudioMorphix and ddim inversion, we set the number of inference steps as 50 while implementing DDPM inversion 200 steps.

## C   QUALITATIVE EVALUATION

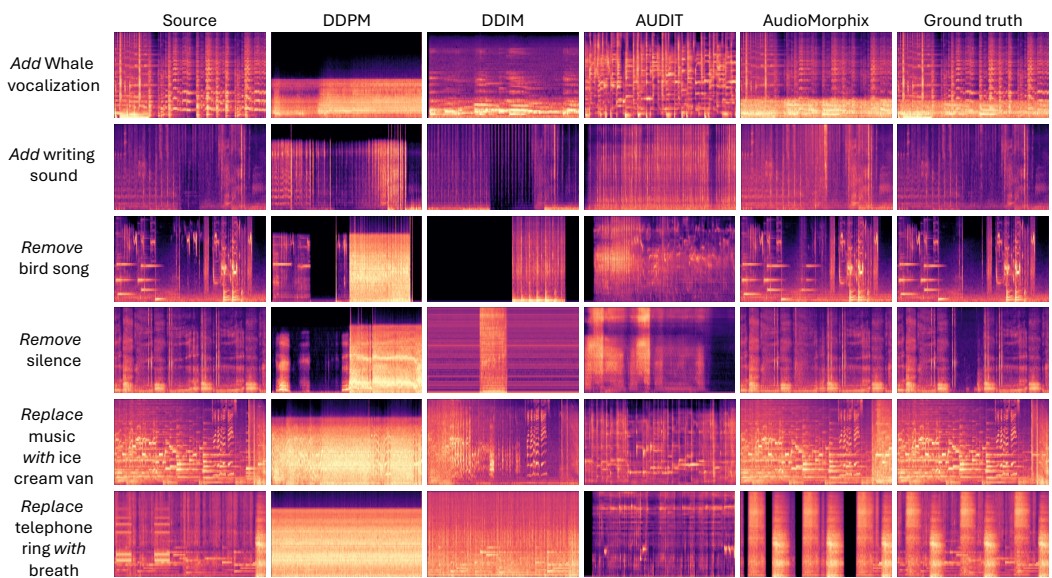

Figure 4: Qualitative evaluation between our AudioMorphix and other audio editing methods.

Figure 4 compares our proposed methods against other audio editing methods, including DDIM, DDPM, and AUDIT, over audio addition, removal, and replacement tasks. It can be observed that our AudioMorphix follows the instructions best. Additionally, the AudioMorphix remains the details of non-targeted region in the raw audio, indicating its capacity of high-fidelity audio editing.

## D   QUALITATIVE EVALUATION

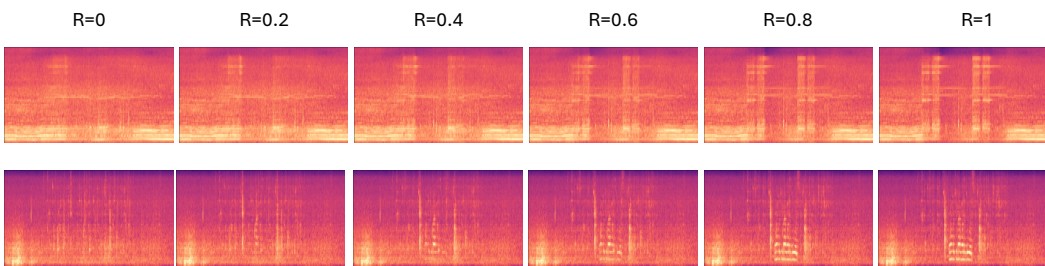

Figure 5: Ablation study on the impact of SLERP in the audio addition task.

Figure 5 indicates the output of the AudioMophix w.r.t. the increase of source-to-reference ratio, the ratio of source audio to the entire mixture. The goal of this experiment is to assess the impact of SLERP operations on the audio addition task. It can be observed that the generated sound smoothly morphed from the source audio to the reference audio. This supports our motivation that a sound mixture can be obtained by morphing between two different sound tracks.

