# OpenReview forum: "AudioMorphix: Training-free audio editing with diffusion probabilistic models"
_ICLR.cc/2025/Conference — ICLR 2025 Conference Withdrawn Submission_

### Official Review · Reviewer_k6nH · 2024-10-28

**Soundness:** 2
**Presentation:** 2
**Contribution:** 2
**Rating:** 3
**Confidence:** 5

**Summary:**

AudioMorphix introduces a novel framework for editing audio without the need for training. The authors conceptualize audio editing as part of a morphing cycle, in which different sounds can be combined into a cohesive audio mixture through morphing, whereas the mixture can be disentangled into individual components via demorphing. They leverage the concept of the audio morphing cycle to optimize the noised latent conditioned on raw input together with reference audio and devise a series of energy functions to refine the guided diffusion process. Additionally, they manipulate the features within self-attention layers to preserve detailed characteristics from the original recordings.

**Strengths:**

1. Technical Depth: The authors delve into the technical details of how AudioMorphix works, including the use of spherical linear interpolation for latent state interpolation and the design of energy functions to guide the diffusion process.
2. New Evaluation Dataset: The creation of AudioSet-E, a new dataset for evaluating audio editing methods, is a valuable contribution to the research community. It provides a standardized way to assess the performance of different audio editing techniques.
3. Broad Applicability: The framework is versatile and can handle a range of audio editing tasks, making it a tool for various applications, from professional audio production to casual users looking to edit audio content.

**Weaknesses:**

1. Confusing charts and tables.
For example, Figure 1 is difficult to read. The layout between the figures is messy. It takes a lot of time to see the relationship between the audio and the processing method. There are still some questions. For example: In Style Transfer, is audio or text needed as a reference signal? The first example of Audio Removal will introduce new content in the edited audio. Is it in line with expectations?

2. Insufficient references and experimental comparisons
There are already published audio editing methods, but there is a lack of references and corresponding comparative experiments. It is not enough to only compare DDIM and DDPM. For example:
[1] Manor H, Michaeli T. Zero-Shot Unsupervised and Text-Based Audio Editing Using DDPM Inversion[C]//Forty-first International Conference on Machine Learning.
[2] Xu M, Li C, Zhang D, et al. Prompt-guided Precise Audio Editing with Diffusion Models[C]//Forty-first International Conference on Machine Learning.

3. Insufficient analysis of experimental results:
Authors need give more analysis about the experimental results. For example, why are the experimental results based on AudioLDM and Tango vary different? Why is Tango better on KL and AudioLDM better on FAD? Why does DDIM have the highest IS among all tasks?
The paper states that "This suggests that AudioMorphix provides more accurate and realistic image edits compared to DDIM inversion, DDPM inversion, and AUDIT methods." However, in fact, in the Addition and Removal tasks, Tango performs worse than DDIM and AUDIT in FAD. In the Replace task, AudioLDM performs worse than DDIM in KL. These experimental results are quite different from the author's conclusion.

4. No subjective evaluation
The objective metrics KL and FAD only measure the feature similarity between the generated audio and the ground truth audio, they cannot measure the quality and audio-text consistency of the audio generation. Audio generation tasks require subjective evaluation. This paper lacks subjective evaluation.

5. Effectiveness of the method
In Figure 1, the addition and removal of audio may add or delete some content at the same time. The experimental results cannot fully prove the effect of the editing, and there is no subjective evaluation. There is no reference audio in the website demo. In summary, it is difficult to prove the effectiveness of this method.

**Questions:**

1. Basic principles of audio editing: validity of edited content and fidelity of unedited content. In Figure 1, the addition and removal of audio may add or delete some content at the same time. Is this reasonable?
For example, in the third picture of Audio Addition, after adding, the content in the original green box disappears. For example, in the first picture of Audio Removal, after deleting, new content is introduced.

2. In the proposed editing methods, is the added/deleted content in audio completely consistent with the reference audio, or is semantic consistency sufficient?

3. If there is mixed audio and original audio, can addition/deletion/replacement be processed directly at the signal level without regeneration? (The experiment did not compare the advantages and disadvantages of this solution with the generated solution.)

4. Using reference audio to edit audio is cumbersome. Has the author thought about the application scenarios of this method? Because audio is a modality that is not easy to visualize. For example: I need to add/delete/replace a certain sound (such as a cat's meow) in the audio. The simplest way is to use text to control it: "Add/delete/replace the cat's meow at 3-6s in this audio". If I construct a "cat's meow at 3-6s" as a reference audio, is it a very troublesome thing (because I need to collect cat's meows and the cat's meows need to be at 3-6s)?

5. Can you further analyze the experimental results?

6. In the webpage you provided, the demo shows only input-audio and ground-truth, with text query but no reference audio. This is what confuses me the most, because your proposed editing is based on the audio reference.

7. The template used in the submitted paper is incorrect.

---

### Official Review · Reviewer_oD6Z · 2024-10-31

**Soundness:** 3
**Presentation:** 1
**Contribution:** 2
**Rating:** 3
**Confidence:** 5

**Summary:**

The paper presented a novel text-to-audio editing method using off-the-shelf pretrained models. New energy functions are introduced to achieve addition and removal with guided diffusion. The system is evaluated in well-known objective metrics. Some audio examples are provided to support the results in tables

**Strengths:**

- The proposed energy function for addition and removal are novel
- Good paper survey + regorous adhoc trials are done to get nice results in objective metrics

**Weaknesses:**

- The novelties of this work are not clearly written. I believe tangent proj. and memory bank are not novel while the proposed energy functions as well as using SLERP are the original contributions of this work. I recommend listing your original contributions in Intro
- The effectiveness of SLERP is not verified. The method should be compared to LERP to confirm your assumption
- The audio samples on the given webpage link for "removal" are not impressive at all, in comparison with the ones on the webpage of AUDIT, which makes me feel skeptical on the demonstrated results in the tables. I'd like to request a small-scale subjective listening test to verify if the objective scores in the tables are correlated with human perceptions
- Some typos: apllied, re-wroten; check the texts in the paper carefully
- "following empirical findings in the image domain" put at least one reference

**Questions:**

- What is x_r^tilda, F_t^c? Not clearly explained
- Why don't you compare your removal function with well-established exisiting methods like target sound extraction, text-queried sound separation?

---

### Official Review · Reviewer_53vA · 2024-11-02

**Soundness:** 1
**Presentation:** 1
**Contribution:** 2
**Rating:** 3
**Confidence:** 4

**Summary:**

This paper introduces a test-time audio editing method called AudioMorphix. It consists in manipulating a target area of a recording using another one as reference. The method relies on regularized latent editing using key/value features manipulation and diffusion guidance. The authors additionally propose a new evaluation benchmark for audio editing.

**Strengths:**

A new dataset (or benchmark?) for audio editing.

**Weaknesses:**

The paper is filled with approximations and expressions that make little sense. The paper refers a lot to previous methods and it is not always clear what is novel and what is inspired from prior work. Some core concepts are not defined (e.g. "energy functions").

**Questions:**

Introduction:
- In the following sentence: "To address the audio transparency, we ...". What does "addressing the audio transparency" mean?
- Is it not contradictory to claim a "training-free framework" relying on "gradient-based optimization"?
- How is "energy functions" defined?
- To what findings does the expression "following empirical findings in the image domain." refer to? Is the key and value components substitution method a contribution or inspired from prior work?

Section 3:
- Region-specific objective: does it make sense to want to keep the rest unchanged during editing? Shouldn't the editing process take into consideration the whole audio so as to keep global consistency and avoid noticeable stitches?
- What does it mean for one pixel to be correlated with more than one sound track?
- What does it mean for a reference audio to be the "interested sound"?
- How do you experimentally observe that the additive property of signals correlates with the latent domain?
- What does "the orthogonal directions of reference audio  ̃xr are not unique in a manifold" mean?
- How are z_c and z_r distributed on a sphere? What sphere?

Section 4:
- What does it mean for a sound track to be "transparent"?

The Figure 1 depicts the task of "style transferring" yet it is defined nowhere.

If this paper proposes a new evaluation benchmark, I would suggest showcasing a few samples from the benchmark.

Typos:
"re-wroten" -> rewritten
"In the community of images" -> in the computer vision community
"AudioMorhix"

---

### Official Review · Reviewer_tr47 · 2024-11-03

**Soundness:** 3
**Presentation:** 2
**Contribution:** 2
**Rating:** 5
**Confidence:** 3

**Summary:**

This paper proposes a novel audio editing method to manipulate raw audio recordings conditioned on reference audio, devising a series of energy functions to refine the diffusion process. The paper also introduces a test set for audio editing.

**Strengths:**

1. This paper explores an interesting topic, which is reference audio-based audio editing.
2. By utilizing a series of energy functions, the method performs better than DDIM inversion.
3. This work introduces a novel test set for audio editing.

**Weaknesses:**

1. [1] and [2] are mentioned in this paper. Zhang's work uses cross-attention control for music editing, while this paper does not compare with this paper in both methodology and experiments. Considering the similarity between these two works, it should be regarded as a weakness of this paper. Additionally, [3] and [4] are related to this work, the authors should include them in the related works.
2. Subjective evaluation is missing. It is always necessary when the audio generation model is proposed since human hearing is still the gold standard in this domain. This paper should consider conducting subjective experiments to prove the editing fidelity and consistency with reference audios like [3].
3. The demo page can still only load very few samples after many attempts. Also, no code link is seen (although the author claims there is one). Pls provide a specific URL for the code repository and suggest ensuring the demo page is fully functional before final submission.
4. The paper mentions using a binary mask for local editing in many places, but it does not explain how this binary mask is constructed. In addition, judging from the few GT audios, such as "remove bird song", that can be displayed on the demo page, these GT audios cannot correspond well to the text queries marked by the author. Pls explain the details of binary mask's construction.
5. Provide the justification for using CLAP without text input, and for an explanation of the rationale behind using different metrics for Audioset-E and MusicDelta.
6. There are some grammatical errors in the paper. For example, re-sample -> re-sampling, proposed -> propose, capabable -> capable.

**Reference**

[1] Hila Manor and Tomer Michaeli. Zero-Shot Unsupervised and Text-Based Audio Editing Using DDPM Inversion, February 2024. URL http://arxiv.org/abs/2402.10009.arXiv:2402.10009 [cs, eess].
[2] Yixiao Zhang, Yukara Ikemiya, Gus Xia, Naoki Murata, Marco Mart´ınez, Wei-Hsiang Liao, Yuki Mitsufuji, and Simon Dixon. MusicMagus: Zero-Shot Text-to-Music Editing via Diffusion Models, February 2024b. URL http://arxiv.org/abs/2402.06178. arXiv:2402.06178 [cs,
eess].
[3] Liu H, Wang J, Huang R, et al. MEDIC: Zero-shot Music Editing with Disentangled Inversion Control[J]. arXiv preprint arXiv:2407.13220, 2024.
[4] Novack Z, McAuley J, Berg-Kirkpatrick T, et al. Ditto: Diffusion inference-time t-optimization for music generation[J]. arXiv preprint arXiv:2401.12179, 2024.

**Questions:**

See questions in weakness.

---

### Official Review · Reviewer_8bhC · 2024-11-04

**Soundness:** 1
**Presentation:** 1
**Contribution:** 2
**Rating:** 3
**Confidence:** 3

**Summary:**

The paper proposes AudioMorphix, as a way for developing training-free audio editing models. Editing is formulated in the form of morphing, where sounds can either be combined to create mixtures or separated – leading to addition, removal and replacement as 3 primary operations. The whole editing process is essentially treated as manipulation of latents for these 3 operations in the morphing cycle. A new dataset AudioSetE is created for experiments.

**Strengths:**

– Training free approach for audio editing is interesting. Done well it could lead to interesting directions for generative methods for audio editing.

– A new dataset is also created. If the dataset is released publicly it would be useful for the community

**Weaknesses:**

– The paper is hard to follow. It’s not clear how the overall framework is coming together. An overall system diagram showing what is going would be helpful. How are all the pieces in the model connected ?

– How will the proposed method handle editing which is a combination of addition, removal and replacement for a given clip ? Also some experimental results on the same are also expected. Moreover, do we really need something extensive to do addition ? Why can't we simply add the signals while maintaining say some SNR ?

– The experimental results and analyses are really weak. Beyond the objective numbers on a new dataset – for which it is hard to fully understand performances, there are no other insights or analyses.  Even some analyses like how useful the energy guidance or the memory bank is, might be helpful.

– The details of the AudiosetE are missing. What are some examples of edit instructions ?

– It’s hard to fully assess these methods without subjective evaluations. CLAP, KLD, FAD do not provide the best assessment of generative audio methods. Some form listening tests or subjective scoring is needed, otherwise it is hard to fully gauge performance improvements.

**Questions:**

Please address the points in weaknesses.

---

### Note · Authors · 2024-11-17

I have read and agree with the venue's withdrawal policy on behalf of myself and my co-authors.